# The effects of verbal and spatial working memory on short- and long-latency sensorimotor circuits in the motor cortex

Markus W. Lenizky, Sean K. Meehan *

Department of Kinesiology and Health Sciences, University of Waterloo, Waterloo, Ontario, Canada

* skmeehan@uwaterloo.ca

## Abstract

Multiple sensorimotor loops converge in the motor cortex to create an adaptable system capable of context-specific sensorimotor control. Afferent inhibition provides a non-invasive tool to investigate the substrates by which procedural and cognitive control processes interact to shape motor corticospinal projections. Varying the transcranial magnetic stimulation properties during afferent inhibition can probe specific sensorimotor circuits that contribute to short- and long-latency periods of inhibition in response to the peripheral stimulation. The current study used short- (SAI) and long-latency (LAI) afferent inhibition to probe the influence of verbal and spatial working memory load on the specific sensorimotor circuits recruited by posterior-anterior (PA) and anterior-posterior (AP) TMS-induced current. Participants completed two sessions where SAI and LAI were assessed during the short-term maintenance of two- or six-item sets of letters (verbal) or stimulus locations (spatial). The only difference between the sessions was the direction of the induced current. PA SAI decreased as the verbal working memory load increased. In contrast, AP SAI was not modulated by verbal working memory load. Visuospatial working memory load did not affect PA or AP SAI. Neither PA LAI nor AP LAI were sensitive to verbal or spatial working memory load. The dissociation of short-latency PA and AP sensorimotor circuits and short- and long-latency PA sensorimotor circuits with increasing verbal working memory load support multiple convergent sensorimotor loops that provide distinct functional information to facilitate context-specific supraspinal control.

## Introduction

Motor performance and learning depend on the efficient integration of sensory afference to generate accurate and appropriate motor commands. Several cortical-cortical and subcortical-cortical sensorimotor loops converge on corticospinal neurons to shape central projections to lower motor neurons and interneurons in the spinal cord [1]. The convergent sensorimotor loops can alter fundamental sensorimotor relationships based on cognitive control processes governing explicit top-down goals [2–5] or implicit processes mediated by subcortical structures like the basal ganglia [6,7] and cerebellum [5,8]. However, the functional significance of different sensorimotor loops remains to be determined.

**Data Availability Statement:** All relevant data are within the manuscript and its Supporting Information files.

**Funding:** Funding for the current work was provided to Sean K. Meehan by the Natural

Sciences and Engineering Research Council of
Canada (RGPIN-2020-04255). The funders had no
role in study design, data collection and analysis,
decision to publish, or preparation of the
manuscript.

**Competing interests:** The authors have declared
that no competing interests exist.

Afferent inhibition is the suppression of a muscular motor response by a sensory afferent volley. It is typically studied by pairing non-invasive electrical nerve stimulation with transcranial magnetic stimulation (TMS) over the primary motor cortex [9–12], providing a tool to quantify the modulatory effect of somatosensory afference converging on output neurons in the motor cortex under varying cognitive and sensorimotor states. Different periods of afferent inhibition can be elicited using both short and long interstimulus intervals. For distal muscles of the upper limb, short-latency afferent inhibition (SAI) occurs when the peripheral electrical stimulus precedes the TMS stimulus by interstimulus intervals of 18–24 ms [9]. The peak inhibitory period of 21–22 ms approximates upper limb somatosensory central conduction. Long-latency afferent inhibition (LAI) occurs across 100–1000 ms interstimulus intervals, with peak inhibition occurring at interstimulus intervals of ~200 ms.

Despite the different interstimulus intervals, SAI and LAI share several similarities. For example, SAI [9] and LAI [13] are cortical in origin. Both reflect the strength of the afferent projection converging on corticospinal neurons in the motor cortex [14,15]. Both are reduced by muscle contraction [16,17]. There is evidence for a degree of somatotopic organization in both the SAI [18–20] and LAI [16] sensorimotor circuits. Pharmacologically, both SAI and LAI are reduced by the γ-aminobutyric acid A (GABA$_A$) receptor agonist lorazepam [21–23] but not agonistic modulation of metabotropic GABA$_B$ receptors by baclofen [23]. The similar response of SAI and LAI to ionotropic GABA$_A$ receptor modulation may reflect a common genesis tied to GABA$_A$ modulation of acetylcholine. SAI is reduced by the muscarinic acetylcholine antagonist scopolamine [24], enhanced by acetylcholinesterase inhibitors [25] and is altered in clinical populations featuring loss of acetylcholine-producing cells, such as Alzheimer's disease [26–28], mild cognitive impairment [26,29] and Parkinson's disease with mild cognitive impairment [30]. To our knowledge, no work has investigated LAI's response to muscarinic or nicotinic modulation. However, LAI is reduced in Parkinson's disease patients despite LAI's insensitivity to dopaminergic medications [31,32]. One theory for reduced LAI in Parkinson's disease is that LAI reductions are driven by a concomitant reduction in cholinergic output [23].

Despite their similarities, less is known about LAI's pathways and functional significance relative to SAI. SAI appears to involve at least one thalamocortical pathway that traverses the primary somatosensory cortex [33,34]. As noted, SAI magnitude is correlated with the amplitude of the N20 evoked potential amplitude [14] that indexes the first arrival of somatosensory afference to primary somatosensory cortex [35]. However, a first order thalamocortical projection to primary somatosensory cortex cannot explain observed dissociations between the N20 somatosensory evoked potential and SAI. For example, SAI can still be elicited despite the absence of the N20 somatosensory evoked potential following a focal lesion of the ventral posterolateral somatosensory relay nucleus of the thalamus [36]. Likewise, SAI can be abolished following a focal lesion of the paramedian thalamus that preserves the first order somatosensory relay nucleus and the N20 somatosensory evoked potential [37]. One theory is that the somatosensory pathway may converge and modulate a direct thalamic projection to the motor cortex [15]. This theory is supported by the reduction, but not complete loss, of SAI following continuous theta burst stimulation to the primary somatosensory cortex [38]. The direct thalamic projection could reflect the bottom-up traits of SAI, like afferent intensity and the somatotopic organization with cortical pathways underlying the malleable traits of SAI.

The malleable traits of SAI include sensitivity to movement goals and cognition. For example, SAI in task-relevant muscles is reduced during movement planning, movement execution and motor imagery [18,39,40]. In contrast, SAI is enhanced in task-irrelevant muscle representations [18,39]. The differential direction of SAI modulation has produced speculation that SAI plays a role in action selection through surround inhibition [41]. Alternatively, the task

relevancy effect across effectors may reflect a movement-related gating phenomena to suppress expected sensory afference and facilitate sensory-based corrections during movement planning and execution [42]. The sensitivity of SAI to GABAergic [23,43] and cholinergic [24] modulation provides a route for cognition to shape sensorimotor processing. For example, the suppression of SAI in task-relevant muscles is mitigated by adopting an internal focus of attention during skilled performance [44] and is consistent with an augmentation of sensory afferent projections from the attended body part [45].

The studies mentioned above exclusively paired peripheral electrical stimulation with monophasic posterior-anterior (PA) TMS. More recently, SAI probed using monophasic anterior-posterior (AP) induced current has defined unique sensorimotor circuits in the motor cortex [4,46–48]. These AP-sensitive circuits have been differentiated from those recruited by PA current given the specific sensitivity of their afferent inputs to cerebellar modulation [5,47] and attention load [4,5]. In contrast, verbal working memory is reported to influence the effect of afferent input to both the PA- and AP-sensitive sensorimotor circuits [3]. While the PA-sensitive circuits are linked to the N20-P27 posterior somatosensory evoked potential generated in the primary somatosensory cortex [35], the AP-sensitive circuits appear to more strongly relate to the frontal P20-N30 somatosensory evoked potential, localized to the supplementary motor area and precentral gyrus [49]. The pattern of responses of the PA and AP sensitive sensorimotor circuits to increasing perceptual attention load [4,5] and increasing verbal working memory load [3] suggests that both circuits share a common modulator but that they also have specific modulators that underlie distinct functional contributions to sensorimotor integration.

Similar obligatory and modulatory pathway organization may also exist in LAI. The limited evidence available suggests that, like SAI, LAI is also modifiable by intrinsic processes. LAI is reduced during muscle contraction of the target muscle [16,17]. One theory Turco et al. [12] put forth is that LAI reflects a later error-driven sensorimotor process. This theory is consistent with evidence that LAI involves a basal ganglia-thalamo-cortical loop [32]. The basal ganglia is critical in regulating kinematic variables during motor performance and error-based motor learning (for a review, see Seidler et al. [50]). Also consistent with this theory, LAI is reduced during early learning, when error-based learning is relatively prominent [51,52]. However, past work has yet to investigate the relationship between cognition and LAI in healthy or clinical populations.

The current study assessed the effects of working memory load on the SAI and LAI circuits recruited by PA- and AP-induced current. Given a common GABA$_A$-mediated mechanism, we hypothesized that SAI and LAI would similarly decrease during memory maintenance as the size of the memory set increased. A secondary purpose investigated the differential effect of visual-verbal versus visuospatial working memory on SAI and LAI sensorimotor circuits. Like verbal working memory, greater visuospatial working memory capacity is predictive of greater rates of explicit [53] and implicit [54] motor sequence learning. Further, the rate of early visuomotor adaptation, another form of motor learning, is also correlated with visuospatial working memory ability [55,56]. The neural structures recruited during visuospatial working memory tasks, particularly the dorsolateral prefrontal cortex, also overlap with those recruited during working memory. Given that the dorsolateral prefrontal cortex plays an integral role in gating afferent projections to the sensorimotor cortex [57], spatial working memory may shape one or both sensorimotor circuits sensitive to PA and AP current. Despite the importance of both verbal and spatial working memory to sensorimotor performance and learning, the effect of visuospatial working memory on SAI and LAI is unexplored.

## Materials and methods

### Participants

Twenty self-reported right-handed adults (6 male, 14 female, 22±2 years, mean±standard deviation) with no history of neurological disease and no contraindications to TMS completed two experimental sessions. All experimental sessions were conducted between July 25th, 2019, to November 18th, 2020. All participants provided written informed consent; the University of Waterloo's Clinical Research Ethics Committee approved the study protocol.

### Verbal working memory task

The verbal working memory task consisted of a delayed match to sample task [58,59]. Participants were seated 70 cm in front of a computer screen. Both arms rested on the work surface with the arms bent at 90°. The left index and middle finger were placed over two adjacent response keys on a Bluetooth number pad.

A trial started with presenting a fixation cross (pre-trial period). Following 500 ms, two or six letters appeared in an array around the fixation cross (Fig 1A). The memory set remained visible for 1500ms, with participants instructed to encode the letters regardless of their locations (encoding period). After 1500ms, the letter set disappeared, and participants had to maintain the memory set for 3000ms (maintenance period). After the maintenance period, a single probe letter was presented in the middle of the screen. Participants indicated whether the probe letter was part of the memory set (probe period) by responding as fast and accurately as possible (S1 Data). The inter-trial period was 3000ms (after the participant's response). If the participant failed to respond within 1500ms, an incorrect response would be recorded, and the subsequent trial would begin following the inter-trial period. Participants failed to respond within 1500ms on less than 1% of all trials.

The memory set was randomly drawn from the twelve letters between A and L. Each letter selected for the memory set appeared at a distinct location around the central fixation cross. There were 12 locations where letters could appear on any given trial. The sites arrayed around two concentric squares centred on the fixation point. The inner square had a width/height of 1.5cm. The outer square had a width/height of 3cm (visual angle, 4.9°). The probability that the probe was drawn from the memory set was 50%, regardless of the memory set size.

### Spatial working memory task

The spatial working memory task was similar to the verbal task [58,59] (S1 Data). The only differences were: 1) a spatial array of dots replaced the spatial array of letters that made up the memory set, 2) the probe consisted of a single dot presented at one of the twelve possible locations, and 3) the participant responded whether the single probe dot was located at one of the locations in the memory set (Fig 1B).

### Transcranial Magnetic Stimulation (TMS)

Consistent with our past work [3], motor-evoked potentials (MEP) were elicited by TMS during the memory maintenance period. The TMS stimulus occurred during the maintenance period (e.g. after the disappearance of the memory set before the presentation of the probe) of the working memory task. The timing of TMS stimulus varied between 1500, 1750, 2000, 2250, or 2500 ms with equal probability using a rectangular distribution. This time range was chosen to place the greatest emphasis on the maintenance processes. The variable timing within the range was used to minimize anticipation of the TMS stimulus.

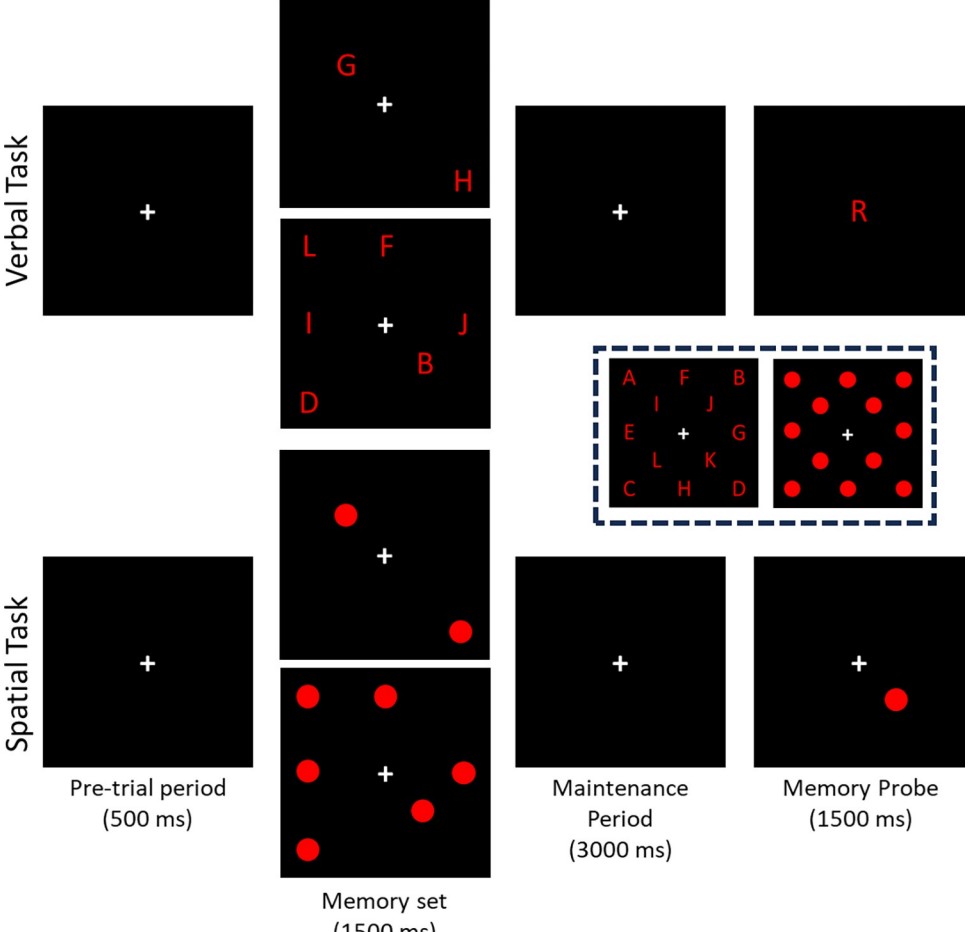

**Fig 1. Examples of the memory set, probes and time courses for the verbal (top) and spatial (bottom) working memory tasks.** The memory set was presented after a 500ms pre-trial period. A group of two or six items appeared. There were twelve locations where an object could appear on a given trial. The inset demonstrates all possible stimuli and locations for the verbal and spatial task. In any single trial, one TMS stimulus alone (unconditioned) or one TMS stimulus preceded by electrical stimulation of the median nerve (conditioned) was delivered during the latter half of the maintenance period. To minimize anticipation of the TMS stimulus, the timing of TMS stimulus varied between 1500, 1750, 2000, 2250, or 2500 ms using a rectangular distribution. The TMS stimulus was triggered by the working memory program using a digital output trigger sent to the digital trigger input on the Magstim 200² stimulator via a BNC connection on the Magstim Stimulator Interface Module. For conditioned trials, a separate digital output trigger was sent to a digital trigger input on the SD9 constant current stimulator. The digital trigger to the peripheral stimulator preceded the digital trigger to the TMS stimulator by either 21 ms or 200 ms.

MEPs were recorded using Labview 2019 software (LabVIEW 2019, National Instruments, Austin, TX) in conjunction with a 2024F 4-channel amplifier (Intronix Technologies Corp., Bolton, ON) and a USB-6341 X Series DAQ (National Instruments, Austin, TX). Surface electromyography electrodes (Ag-AgCl) were placed over the first dorsal interosseous muscle. The active electrode was placed over the muscle belly of the first dorsal interosseous muscle. The reference electrode was placed over the tendon at the base of the second phalange. Surface electromyography recordings were performed as epochs that started 0.3 s prior to the TMS stimulus and ended 0.5 seconds after the TMS stimulus. The recording of each epoch was triggered by a 5V TTL pulse sent from a digital output on the TMS stimulator to a digital input on the USB-6341 DAQ connected to the 4 channel electromyographic amplifier. During acquisition, data were amplified (x1000), digitized (x4000Hz) and filtered (band pass filtered 5-1000Hz,

notch filter– 60Hz). The MEP was defined as the peak-to-peak amplitude of the maximal electromyography response between 20 to 50ms post-TMS stimulation (S1 Data). Trials were excluded from subsequent analysis if the root mean square error of the electromyographic signal from the first dorsal interosseous muscle exceeded 15μV during the pre-TMS stimulus interval (-50 to 0ms). Less than 1% of all trials were excluded because of excessive muscle activity during the pre-TMS stimulus interval.

TMS was delivered using a Magstim 200$^2$ stimulator (Magstim Company Ltd, Wales, UK) and a 50mm figure-8 branding coil (D50 Alpha B.I., Magstim Company Ltd, Wales, UK). The coil was placed tangential to the scalp at 45˚ to the midline over the left motor cortex. Two different current directions were employed. Posterior-anterior (PA) current was induced in the underlying tissue by orienting the coil so the current flow in the coil moved from anterior to posterior. To induce anterior-posterior (AP) current in the underlying tissue, the coil was rotated 180˚.

The left FDI motor cortical hotspot was defined as the scalp position that elicited the largest and most consistent response following PA stimulation. The location and trajectory of the coil on the scalp at the hotspot were recorded using the BrainSight™ stereotactic system (Rogue Research, Montreal, Québec, Canada). Consistent with our past work [3–5], the same hotspot was used for AP stimulation [60].

Test stimulus intensity was determined separately for the PA and AP-induced current using the maximum likelihood parameter estimation by sequential tracking (ML-PEST) adaptive threshold-hunting method [61]. The ML-PEST method uses a binary, yes, or no response to model an S-shaped function of the probabilistic nature of evoking an MEP above the 1mV peak-to-peak threshold at a given stimulus intensity. For nine participants, an MEP greater than 1mv was not achieved with AP-induced current before reaching 100% maximal stimulator output. In such cases, the ML-PEST threshold method was repeated with a test stimulus intensity target of 0.750mV and, if necessary, 0.5mV. If an AP MEP of at least 0.5mV could not be elicited, then the participant only completed the part of the procedure involving PA-induced current. A minimum MEP of 0.5mV was used as the magnitude of SAI is consistent across MEPs ranging from 0.5 to 2mV regardless of induced current direction [46].

## Afferent inhibition

Afferent inhibition consisted of a peripheral electrical stimulus paired with TMS. Electrical stimulation was delivered using an SD9 constant current high voltage stimulator (Grass Astro-Med, West Warwick, RI). Stimulation was applied over the median nerve at the right wrist (constant current square wave pulse, 0.2 ms duration, cathode proximal). Electrical stimulation intensity was set to yield a slight thumb twitch (~0.2 mV APB M-wave). For SAI, the conditioning electrical stimulation preceded TMS by 21 ms. For LAI, conditioning electrical stimulation preceded TMS by 200 ms. The magnitude of SAI and LAI were derived by expressing the MEP amplitude evoked by TMS stimuli conditioned by peripheral electrical stimulation (conditioned) as a percentage of the MEP amplitude elicited by TMS alone (unconditioned) within each task variant, trial type and TMS current.

$$\% \text{ SAI or LAI} = \frac{\text{Conditioned } MEP}{Unconditioned\ MEP} x\ 100$$

## Experimental design

The experiment consisted of two similar sessions during which TMS procedures were identical. The memory tasks were separated across the two sessions so that in one session, participants only performed the verbal variant of the delayed match to sample task while SAI/LAI

were quantified. In the other session, participants only performed the spatial variant of the delayed match to sample task while SAI/LAI were quantified. The order of the sessions was randomized across participants. Within each session, SAI and LAI were assessed using TMS that induced either PA or AP current. Participants completed all trials for one induced current before completing the trials for the other induced current. The order of induced current direction was randomized within the spatial/verbal order of the task so that equal numbers of participants completed all combinations of task and induced current order. For example, if a participant completed the verbal session first and within that session completed the PA induced current trials before the AP induced current trials then they also completed the PA induced current trials before the AP induced current trials for the spatial session.

Within each session participants completed 240 trials of the given task. One hundred twenty trials involved TMS that induced PA current. The remaining 120 trials involved AP-induced current. The 120 trials were broken into 12 blocks with ten trials per block, one TMS stimulus per trial. Each block consisted of either unconditioned TMS (TS Alone), median nerve stimuli that preceded the TMS stimulus by 21ms (SAI), or median nerve stimuli that preceded the TMS stimulus by 200ms (LAI). Within a block, half occurred during a working memory task trial with a memory set size of two items. The remaining half involved six items. The order of the blocks followed a snake pattern (LAI-TS Alone-SAI-SAI-TS Alone-LAI) that was repeated twice, yielding 20 unconditioned stimuli, 20 SAI trials and 20 LAI trials for each set size and induced current.

For the four participants where an AP MEP of at least 0.5mV was not elicited, only the 120 trials involving PA-induced current were completed during the verbal and spatial sessions.

## Data analysis

Statistical analyses were performed using the R environment for statistical computing (version 3.6.1) [62]. The following packages were used: "rstatix" [63] and "tidyverse" [64]. The Shaprio-Wilk test and Q-Q plots were used to assess the normality of the distributions.

The behavioural and physiological analyses were conducted separately for the PA and AP current directions.

Response time and accuracy failed to meet the normality assumption for both PA and AP current trials. Therefore, the Aligned Rank Transform tool (ARTool) for R [65] was used to perform separate non-parametric equivalents of a Task (Verbal, Spatial) x Set Size (Two-Items, Six-Items) repeated measures ANOVA for each current direction.

For SAI and LAI, separate parametric Task (Verbal, Spatial) x Set Size (Two-Items, Six-Items), repeated measures ANOVAs were conducted. Significant two-way interactions were decomposed using the simple main effect of Task.

## Results

### Behaviour

Table 1 illustrates the mean response time and accuracy for each task, current direction and set size.

The non-parametric Task x Set Size repeated measures ANOVA on response time revealed significant main effects of Task [$F_{1,54} = 14.28$, p<0.00039] and Set Size [$F_{1,54} = 53.48$, p<0.00001] as well as a Task x Set Size interaction [$F_{1,54} = 4.58$, p<0.036] for PA current. The corresponding analysis for response time for AP current revealed significant main effects of Task [$F_{1,42} = 18.48$, p = 0.0001] and Set Size [$F_{1,42} = 29.21$, p<0.00001]. The interaction was not significant [$F_{1,42} = 2.47$, p<0.12]. For both PA and AP current, response time was slower for the six-item set than for the two-item set. For both PA and AP current, response times

**Table 1. Response time and accuracy (mean±standard error) for the two- and six-item set sizes of the verbal and spatial tasks.**

| | Response Time (ms) | | Accuracy (%) | |
|---|---|---|---|---|
| Posterior-Anterior (PA) | Verbal | Spatial | Verbal | Spatial |
| Two-Items | 864 (23) | 789 (29) | 90 (3) | 92 (3) |
| Six-Items | 932 (27) | 908 (29) | 86 (2) | 85 (2) |
| Anterior-Posterior (AP) | | | | |
| Two-Items | 911 (33) | 804 (22) | 93 (1) | 95 (2) |
| Six-Items | 973 (34) | 916 (26) | 88 (2) | 86 (2) |

were slower for the verbal compared to the spatial task. The Task x Set Size interaction for PA current was driven by the convergence in response time across the verbal and spatial tasks from two-items (contrast, p = 0.0007) to six-items (p = 0.68). A similar convergence pattern was seen for the AP current. However, the convergence was marginally weaker.

The non-parametric Task x Set Size repeated measures ANOVA on accuracy revealed a main effect of Set Size [$F_{1,54}$ = 33.65, p<0.00001] for PA current. Accuracy was lower on six-item set size trials than on two-item trials (two-items = 91±2%, six-items = 85±2%). The main effect of Task [$F_{1,54}$ = 0.06, p = 0.80] and the Task x Set Size [$F_{1,54}$ = 1.95, p = 0.17] interactions were not significant.

The corresponding analysis for AP current similarly revealed a main effect of Set Size [$F_{1,42}$ = 23.54, p = 0.00002]. Accuracy was lower on six-item set size trials compared to two-item set size trials (two-items = 94±1%, six-items = 87±2%). Again, the main effect of Task [$F_{1,42}$ = 0.30, p<0.59] and the Task x Set Size interaction [$F_{1,42}$ = 2.83, p<0.10] were not significant.

## Short-Latency Afferent Inhibition (SAI)

The individual data and group averages for PA- and AP-SAI across the verbal and spatial tasks are shown in Fig 2.

The two-way repeated-measures ANOVA on PA SAI revealed a significant Task x Set Size interaction [$F_{1,19}$ = 4.63, p = 0.04, $\eta_p^2$ = 0.20]. The interaction was driven by a reduction in PA SAI from two-items to six-items for the verbal task (p = 0.02) but no difference for the spatial task (p = 0.99) (Fig 2B, left panel).

The corresponding analysis for AP SAI failed to reveal a significant Task x Set Size interaction [$F_{1,15}$ = 0.07, p = 0.80, $\eta_p^2$ = 0.004]. Neither the main effects of Task [$F_{1,15}$ = 0.37, p = 0.55, $\eta_p^2$ = 0.02] or Set Size [$F_{1,15}$ = 0.47, p = 0.50, $\eta_p^2$ = 0.03] reached significance (Fig 2B, right panel).

## Long-Latency Afferent Inhibition (LAI)

The individual and group averages for PA- and AP-LAI across the verbal and spatial tasks are shown in Fig 3.

The two-way repeated-measures ANOVA on PA LAI failed to reveal a significant Task x Set Size interaction [$F_{1,19}$ = 0.85, p = 0.37, $\eta_p^2$ = 0.04]. Neither the main effect of Task [$F_{1,19}$ = 0.26, p = 0.62, $\eta_p^2$ = 0.01] or Set Size [$F_{1,19}$ = 0.50, p = 0.49, $\eta_p^2$ = 0.26] reached significance (Fig 3B, left panel).

The corresponding analysis for AP LAI failed to reveal a significant Task x Set Size interaction [$F_{1,15}$ = 1.15, p = 0.30, $\eta_p^2$ = 0.07]. Neither the main effects of Task [$F_{1,15}$ = 0.37, p = 0.55, $\eta_p^2$ = 0.02] or Load [$F_{1,15}$ = 0.04, p = 0.84, $\eta_p^2$ = 0.003] reached significance (Fig 3B, right panel).

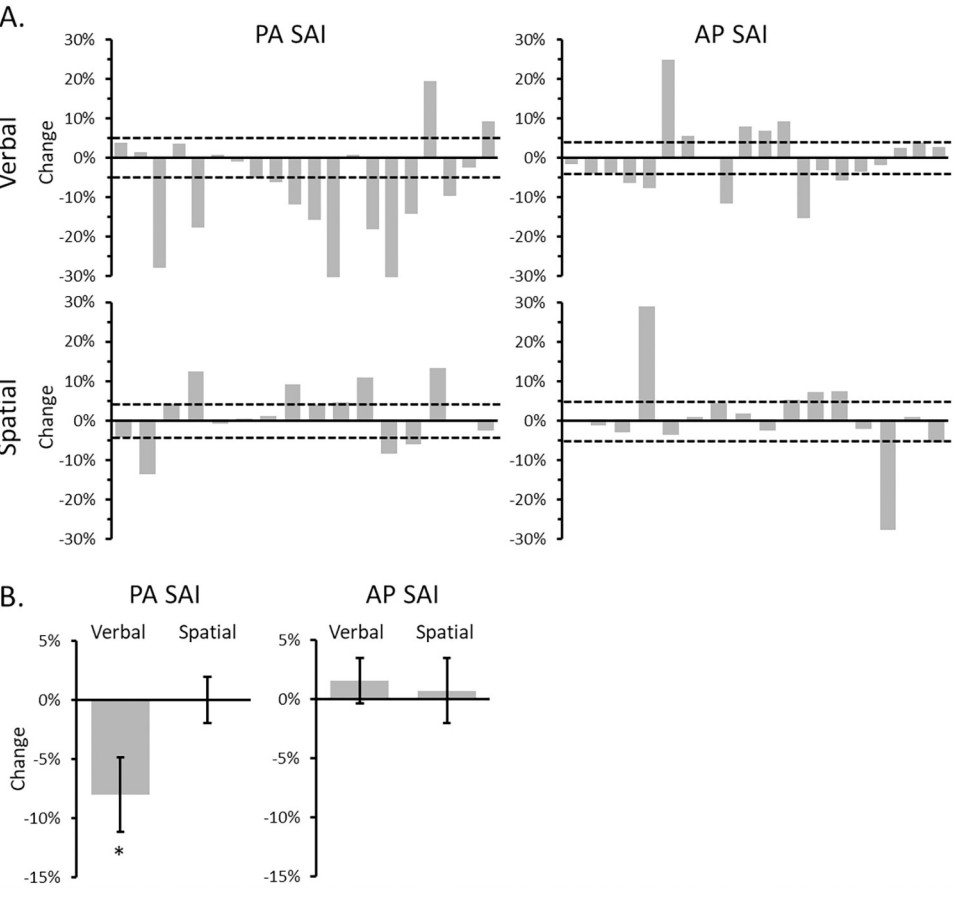

**Fig 2. The change in SAI for each participant and the group average.** (A) The individual change in SAI is illustrated for SAI elicited using PA-induced current during the verbal task (top left), PA-induced current during the spatial task (bottom left), AP-induced current during the verbal task (top right) and AP-induced current during the spatial task (bottom right). The change in SAI was calculated by subtracting the magnitude of SAI during six-item maintenance from the magnitude during two-item maintenance. Negative values indicate more inhibition during two-item trials. Positive numbers indicate more inhibition during the six-item trials. The horizontal dashed lines represent the change corresponding to a moderate effect (Cohen's $d = 0.5$). (B) The group average change in PA SAI (left) and AP SAI (right) for the verbal and spatial tasks. Error bars indicate the standard error of the mean. *Denotes a significant contrast between two- and six-items.

## Discussion

The current study assessed the effect of visual-verbal and visuospatial working memory load on the SAI and LAI sensorimotor circuits recruited by PA- and AP-induced TMS current. PA SAI, but not AP SAI, decreased with the requirement to maintain a larger number of items in the memory set. Although we were able to elicit LAI using both PA- and AP-induced current, neither PA LAI nor AP LAI was sensitive to verbal memory maintenance demands. Neither SAI nor LAI elicited using PA- and AP-induced currents were sensitive to the number of visuospatial locations to be maintained for subsequent recall.

The current study is the first to assess the effect of working memory on LAI. The reduction in PA SAI during maintenance of an increasing verbal memory set size replicates our past work using a similar numeric verbal working memory task [3]. The insensitivity of PA LAI to increasing working memory load is quite interesting. At rest, PA SAI and PA LAI share common pharmacology [23], and their response to the amplitude of the afferent volley is similar

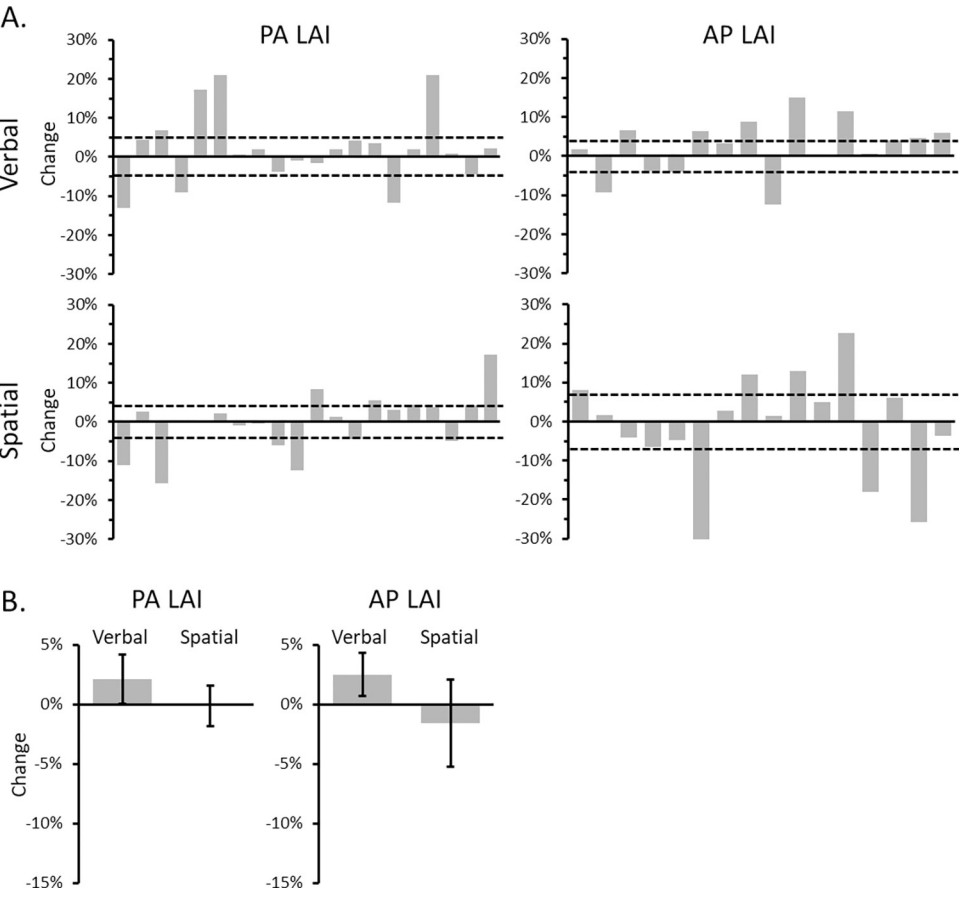

**Fig 3. The change in LAI for each participant and the group average.** (A) The individual change in LAI is illustrated for LAI elicited using PA-induced current during the verbal task (top left), PA-induced current during the spatial task (bottom left), AP-induced current during the verbal task (top right) and AP-induced current during the spatial task (bottom right). The change in LAI was calculated by subtracting the magnitude of LAI during six-item maintenance from the magnitude of LAI during two-item maintenance. Negative values indicate more inhibition during two-item trials. Positive numbers indicate more inhibition during the six-item trials. The horizontal dashed lines represent the change corresponding to a moderate effect (Cohen's $d$ = 0.5). (B) The group average change in PA LAI (left) and AP LAI (right) for the verbal and spatial tasks. Error bars indicate the standard error of the mean.

[14,15]. PA SAI and PA LAI also show similar movement-related gating during simple ballistic tasks with minimal movement accuracy requirements [16,17]. The similarities across PA SAI and PA LAI suggest a common early locus of afferent modulation. The longest likely pathway involved in PA SAI is a thalamo-primary somatosensory-primary motor cortex projection [34]. The similarities across PA SAI and PA LAI are consistent with a common thalamocortical locus of modulation. For PA LAI, the effect of this early modulation is then projected to subsequent parts of the LAI pathway. The insensitivity of PA LAI to increasing verbal working memory load in the current study is not consistent with a single site of modulation. The early reduction of afference, reflected by the reduction in PA SAI, should have been reflected in the PA LAI pathway. Instead, the insensitivity of PA LAI to verbal working memory is consistent with an early suppression of the thalamocortical projections followed by an enhancement of the afferent projection upstream. The former would account for the similar pharmacology and exogenous responses of PA SAI and PA LAI, as well as the dissociation of PA SAI and PA LAI to increased verbal working memory load.

A second modulation site is also consistent with the dissociation of PA SAI and PA LAI following the initial practice of a complex motor skill [51]. The 200 ms inter-stimulus interval used to elicit LAI leaves substantial time for the afferent information to traverse pathways involving the bilateral secondary somatosensory areas in the inferior parietal lobe, association areas in the posterior parietal cortex, prefrontal and premotor areas [66] as well as the basal ganglia [32]. The left inferior parietal lobe and premotor motor areas are activated during sub-vocal rehearsal [67]. Convergence of the LAI afferent projection with the phonological representations of the memory set could lead to the facilitation of the somatosensory afference, especially under a high working memory load. Such an explanation is consistent with behavioural studies of cognitive control loading. The ability to suppress task-irrelevant afferent information, breaks down when cognitive control functions are loaded, leading to increased distraction [68]. The high degree of heterogeneity in PA LAI response to both verbal and spatial working memory may reflect interindividual variability in the nature of the upstream process(es) that shape the afferent input as it traverses various nodes in the afferent pathways over the 200 ms before it converges in primary somatosensory cortex. In contrast, the relatively limited time window for SAI may lead to more homogeneity in the effect of working memory on the shorter latency thalamocortical projections. This is supported by the individual data where moderate to large reductions in PA SAI during verbal working memory were more consistently elicited across participant.

The dissociation of PA SAI and AP SAI during verbal working memory maintenance is surprising, given past work demonstrating a similar sensitivity of these sensorimotor circuits to working memory demands [3]. Suzuki and Meehan [3] hypothesized that the common effect of verbal working memory load across induced current reflected the effect of working memory on basket cells in layer IV of the motor cortex. Basket cells have inhibitory projections to the perisomatic region of the corticospinal neuron. Basket cells receive excitatory afferent projections from somatosensory pyramidal neurons and thalamus [69] and, similar to SAI, are sensitive to cholinergic modulation [70]. The theory was the concomitant effect observed by Suzuki and Meehan [3] was driven by a reduced excitation of the basket cells that depolarized the perisomatic region of the corticospinal neuron and shunted more distal inputs from the AP SAI pathway [71]. An alternate theory is that the common suppression of PA and AP SAI is mediated by a third party, such as LICI [12]. LICI is also sensitive to increasing verbal working memory loads [72] and can inhibit PA SAI [73]. However, the dissociation of verbal working memory on PA and AP SAI circuits in the current study is inconsistent with either hypothesis.

An alternate explanation for the PA SAI-AP SAI dissociation is the enhanced experimental control over the perceptual elements of the working memory task in the current study. AP, but not PA SAI, is sensitive to attention load [4,5]. The task used by Suzuki and Meehan [3] consisted of numeric memory sets of two and six items with an equal probability that the probe would be drawn from the original memory set. However, there were only nine possible numbers to draw the probe from. Although the participant may have committed to subvocally rehearsing the entire memory set, the perceptual attention demands of the task may have also changed with increasing memory load. This possibility is supported by concurrent changes in the parietal N20-P27 and frontal P20-N30 somatosensory evoked potentials with the increased set size observed by Suzuki and Meehan [3]. Mirdamadi et al. [4] have previously linked AP SAI, and the frontal P20-N30 somatosensory evoked potential through manipulation of perceptual attention load. The current study better controlled for perceptual influences by having twice as many possible probe letters than the largest memory set, reducing the possibility that perceptual attention load concurrently changed with working memory load.

The dissociation of PA SAI across verbal and spatial delayed match to sample tasks suggests that verbal and spatial working memory differentially impact sensorimotor integration. A

similar dissociation has been observed across spatial and non-spatial attention. Kotb et al. [45] reported enhanced PA SAI when attention was directed towards the hand contralateral to the cortex in which SAI was quantified. In contrast, PA SAI is not influenced by the perceptual demands of a non-spatial task despite the sensitivity of AP SAI circuits to the same task [4,5]. If the PA SAI circuits are primarily involved in movement planning, the absence of any effect of visuospatial working memory likely reflects the absence of a spatial attention component centred upon the body.

One limitation of the current study is that SAI was only assessed in the dominant left hemisphere. The consensus is that verbal and spatial working memory are lateralized to the left and right hemispheres, respectively [74,75]. The effect of visuospatial working memory on PA SAI, and perhaps LAI, may be muted by assessing afferent inhibition in the dominant left hemisphere. However, a lateralization explanation seems less likely. Increased working memory demands using a similar task are associated with reduced lateralization [59]. The increased recruitment of the opposite hemisphere with increasing spatial working memory load should have had some effect on afferent inhibition assessed in the left hemisphere. Further, past work demonstrates strong relationships between spatial working memory and sensorimotor ability in the dominant hemisphere [53–56]. One final explanation that cannot be ruled out is that the insensitivity of afferent inhibition to visuospatial working memory load could reflect differential encoding strategies if participants represented the spatial array as a singular object. The subsequent presentation of the probe would be less about recall than recognition, driven by a comparison of the maintained visuospatial stimulus representation and the probe. The slower response times and decreased accuracy, consistent with past work using a similar task [58], may reflect slower processing speed of the more complex visual stimulus during memory scanning following probe presentation rather than load during the maintenance period. Therefore, we cannot rule out the possibility that the load manipulation of the visuospatial task failed to manipulate the load during memory maintenance as extensively across individuals as the verbal working memory manipulation. In general, the spatial working memory task yielded weaker change in PA SAI in most individuals that was randomly distributed around zero. In contrast, the verbal working memory task demonstrated stronger and more consistent reductions in PA SAI across individuals. Variation in visuospatial maintenance strategies may explain the random distribution of the weak directional changes around zero. Future work should employ a visuospatial manipulation where the spatial information is not presented as a static display to delineate the sensitivity of sensorimotor pathways to visuospatial working memory in simple static versus complex dynamic environments. For example, the memory set could be a sequence of locations to necessitate individual storage of each spatial bit of information, but that cannot be easily verbalized. However, the challenge of sequential memory tasks, like the n-back is that memory maintenance may be confounded by encoding of subsequent elements of the memory set.

The current study has additional limitations that should be acknowledged. One limitation is that the MagStim $200^2$ stimulator has a pulse duration of ~82 μs. This pulse duration recruits a mix of unique AP-sensitive inputs to the cortical spinal neuron [76] that may have different functional properties [47]. The discrepancy between the effect of verbal working memory on AP SAI in the current study and Suzuki et al. [3], may reflect the differential propensity to recruit $AP_{30}$ vs. $AP_{120}$ sensorimotor circuits with a hybrid pulse width of 82 μs. $AP_{30}$ current recruits sensorimotor circuits with longer MEP onset latencies [47,76] and sensitivity to cerebellar modulation [47]. In contrast, $AP_{120}$ current recruits sensorimotor circuits with shorter MEP onsets closer to those of $PA_{30}$ and $PA_{120}$ sensorimotor circuits. Therefore, our assessment of AP SAI and AP LAI may reflect a different proportion of heterogeneous recruitment of AP circuits. This issue appears less prominent for PA current as the available evidence for $PA_{30}$

and PA$_{120}$ shows similar MEP onset latencies and functional properties [47,76]. Future work should employ controllable pulse parameter TMS stimulation to better isolate the different AP (and potentially PA) sensitive circuits by using different TMS pulse configurations.

A final limitation of the current study was the averaging of TMS stimuli over a limited window within the maintenance period of the experimental task at which SAI and LAI were assessed. The timing of the SAI assessment during the memory maintenance window was set later in the window to emphasize mental rehearsal. The sensitivity of the different sensorimotor circuits may change from early to late rehearsal during the maintenance phase. Unfortunately, we did not track the specific timing of each trial or have sufficient trials to investigate the dynamic effect of working memory. Future work should consider that afferent inhibition is a dynamic phenomenon.

## Conclusion

The current study is the first to assess the effect of cognition on LAI. In contrast to SAI, LAI is not consistently influenced by either verbal or visuospatial working memory. The dissociation of PA SAI and PA LAI suggests that verbal working memory modulates sensory afference early in the somatosensory processing stream but that interactions later in the processing stream may mitigate the utility of LAI to be used as a marker of cognitive influence over sensorimotor integration.

## Supporting information

**S1 Data.**
(XLSX)

## Acknowledgments

The authors would like to thank Larysa Martin and Braeden Hof for their help during data collection.

## Author Contributions

**Conceptualization:** Markus W. Lenizky, Sean K. Meehan.

**Formal analysis:** Markus W. Lenizky.

**Funding acquisition:** Sean K. Meehan.

**Investigation:** Markus W. Lenizky.

**Methodology:** Markus W. Lenizky, Sean K. Meehan.

**Software:** Sean K. Meehan.

**Supervision:** Sean K. Meehan.

**Visualization:** Markus W. Lenizky.

**Writing – original draft:** Markus W. Lenizky, Sean K. Meehan.

**Writing – review & editing:** Markus W. Lenizky, Sean K. Meehan.

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
