## [Decision Letter · Decision Letter 0]

7 Feb 2024

PONE-D-23-40285The effects of verbal and spatial working memory on short- and long-latency sensorimotor circuits in the motor cortexPLOS ONE

Dear Dr. Meehan,

Thank you for submitting your manuscript to PLOS ONE. After careful consideration, we feel that it has merit but does not fully meet PLOS ONE’s publication criteria as it currently stands. Therefore, we invite you to submit a revised version of the manuscript that addresses the points raised during the review process.

Two expert reviewers evaluated your manuscript. Both reviewers find your paper interesting and worthwhile, but they also raise concerns regarding the data analyses. These relate to possible effects of the different TMS interval on the reported data. Given the specific comments below, I think it should be possible to fully address all of the reviewer concerns, so I look forward to receiving the revised version of your work.

We look forward to receiving your revised manuscript.

Kind regards,

Dimitris Voudouris

Academic Editor

PLOS ONE

“Funding provided to Sean K. Meehan by the Natural Sciences and Engineering Research Council of Canada (RGPIN-2020-04255)”

Reviewers' comments:

Reviewer's Responses to Questions

**Comments to the Author**

1. Is the manuscript technically sound, and do the data support the conclusions?

Reviewer #1: Yes

Reviewer #2: Yes

2. Has the statistical analysis been performed appropriately and rigorously? 

Reviewer #1: Yes

Reviewer #2: Yes

3. Have the authors made all data underlying the findings in their manuscript fully available?

Reviewer #1: Yes

Reviewer #2: Yes

4. Is the manuscript presented in an intelligible fashion and written in standard English?

Reviewer #1: Yes

Reviewer #2: Yes

5. Review Comments to the Author

Reviewer #1: The authors report short and long -latency afferent inhibition to probe the influence of verbal and spatial working memory load. They describe that specific sensorimotor circuits recruit, with different working memory load and probed short and long -latency afferent inhibition by PA and AP TMS-induced current.

However, there are some issues to be addressed.

1. It is not clear how the authors consider TMS stimuli delivery period (i.e 1500, 1750, 2000, 2250 and 2500) during analysis. I see every conditioned was averaged. It is not clear what is the effect on SAI and LAI with different TMS stimuli delivery period. It was also not clear how MEP elicited during these different TMS stimuli delivery period.

2. It is difficult to understand unconditioned or conditioned TMS stimuli, there is no clear statement.

3. It is not clear that why the authors called the task as verbal working memory, it is set of different character (shape).

4. It will be good if authors explain the interstimulus intervals (ISIs) in line 53.

5. Is there any study suggesting that SAI appears to involve a thalamic projection to the motor cortex (line 75)?

6. Please give some reference to support the dissociation between perceptual load and working memory demands suggesting some common elements across the different SAI sensorimotor circuits.

7. The line number 179 is not clear.

8. Does PA in the verbal task reflect subtle differences in SAI and not in AP (SAI as well as LAI)?

9. The authors mentioned every critical question to be asked in this article as limitation of this paper.

Reviewer #2: This study examined the response of short (SAI) and long-latency afferent inhibition (LAI) to verbal and spatial working memory loads using posterior-anterior (PA) and anterior-posterior (AP) TMS-induced currents. Participants maintained sets of two or six verbal or spatial stimulus items in two sessions, differing only in current direction. PA SAI decreased with verbal load, while AP SAI remained unchanged. Visuospatial load did not affect PA or AP SAI. Neither PA nor AP LAI responded to memory load. The authors suggest that distinct sensorimotor circuits facilitate context-specific supraspinal control amid increasing verbal working memory load.

The authors have described their methodology well and the manuscript is clear and concise. However, I have one minor comment.

1. I think the authors should make some reference to the work using spatial working memory, as it is not clear why they have chosen this task to compare with verbal working memory.

2. The order of the trials with AP and PA-induces current is not clear in the methodology. Can the authors say whether they followed an order or whether it was randomised?

3. Seeing that there are differences in the TRs, I suggest that the authors could carry out the analyses they have done by introducing the TR as a covariate, so as to shed some more light on the results obtained.

4. In both Figure 2 and Figure 3 there is a lot of variability in the effects between participants. Can you address this issue in the discussion?

5. Could you produce a figure showing only the participant averages for both SAI and LAI?

6. PLOS authors have the option to publish the peer review history of their article (what does this mean?). If published, this will include your full peer review and any attached files.

Reviewer #1: **Yes: **Balbir Singh

Reviewer #2: No

---

## [Author Response · Author response to Decision Letter 0]

22 Mar 2024

Response to Reviewer Comments

We thank the reviewers for their insightful comments on the manuscript. We have addressed each point and made the applicable revisions. Below, we have enumerated their specific comments and replied to each. Changes made to the manuscript text are highlighted in red. In our responses, specific line numbers reference the revised manuscript. 

Reviewer #1 

Point 1. It is not clear how the authors consider TMS stimuli delivery period (i.e 1500, 1750, 2000, 2250 and 2500) during analysis. I see every conditioned was averaged. It is not clear what is the effect on SAI and LAI with different TMS stimuli delivery period. It was also not clear how MEP elicited during these different TMS stimuli delivery period.

Response 1. We apologize for any confusion. We have revised the text in the Figure 1 caption (lines 162 to 171) and lines 187 to 192 to clarify the MEP acquisition process. Briefly, a single TMS stimulus was delivered for each trial, either 1500, 1750, 2000 or 2250 after the memory set disappeared but before the probe appeared. We varied the timing of the TMS stimulus to minimize anticipation of the TMS stimulus. The TMS stimulus on a given trial was either TMS delivered without electrical median nerve stimulation (unconditioned) or the same TMS stimulus preceded by electrical stimulation of the median nerve (conditioned). The probability of the TMS stimulus occurring at any given time on a single trial was equiprobable (controlled by a rectangular distribution). 

To get SAI, we divided the average of all conditioned stimuli delivered across the 1500, 1750, 2000, 2250 and 2500 time points by the average of all unconditioned stimuli delivered across the 1500, 1750, 2000, 2250 and 2500 time points for a given set size and current direction. As acknowledged in the limitations section, we do not have sufficient trials at each specific time point to systematically investigate changes in SAI/LAI across the 1500-2500 ms of the maintenance period. There was not enough time to collect sufficient trials to calculate SAI reliably at each time point.

Point 2. It is difficult to understand unconditioned or conditioned TMS stimuli, there is no clear statement.

Response 2. We apologize for the ambiguity. We have revised text in the introduction (lines 41 to 47), the methods (lines 241 to 243) and the Figure 1 caption (lines 162 to 163) to make clear that “conditioned” refers to MEPs elicited by TMS stimuli that were preceded by peripheral electrical stimulation and that “unconditioned” refers to TMS stimuli that were not preceded by peripheral electrical stimulation. 

Point 3. It is not clear that why the authors called the task as verbal working memory, it is set of different character (shape).

Response 3. Our terminology is consistent with the cited neuroimaging work using similar verbal and spatial Sternberg tasks to dissociate the neural substrates of verbal and spatial working memory (Thomason et al. J Cog Neurosci 2009; Reuter-Lorenz et al. Neuroscience 2000; Smith et al. Proc Natl Acad Sci 1996). These studies and many others use letters or digits to engage the verbal working memory system, which is responsible for temporarily storing letters, digits, words, and sentences over several seconds.

We agree that letters, digits, and even words or sentences may be matched to a probe based on shape in specific circumstances. For example, chronometric studies of stimulus classification often rapidly present a letter or word probe adjacent to or immediately after the memory set to promote recognition based on shape. Given the strong association between letter shape and name, increasing the delay between the memory set and the probe provides time for processing the stimuli at deeper levels to facilitate subvocal rehearsal. 

Given the success of the cited neuroimaging work in dissociating verbal and spatial neural substrates with similar maintenance periods (~3 seconds) and that we conducted our TMS assessments during the maintenance period (before probe presentation at the central location), we are confident that we are probing the active rehearsal of verbal information even if part of the matching process following probe presentation may be based on recognizing a specific letter shape. 

Point 4. It will be good if authors explain the interstimulus intervals (ISIs) in line 53.

Response 4. We apologize for the oversight in defining the abbreviation ISI on first use in the original manuscript. We have removed the abbreviation from the revised manuscript completely. Further, we have revised lines 44-47 to explain better the inter-stimulus interval in the context of afferent inhibition.

Point 5. Is there any study suggesting that SAI appears to involve a thalamic projection to the motor cortex (line 75)?

Response 5. We appreciate that there is no definitive evidence to support a direct thalamic projection to the motor cortex in the generation of SAI. However, there is evidence that dissociates the N20 somatosensory evoked potential and SAI in response to thalamic lesions affecting the primary thalamic relay nuclei projecting to S1 and SAI, supporting the involvement of another relatively direct influence given the tight time window.

We have revised the text (lines 68 to 79) to illustrate better the evidence for a thalamocortical pathway that traverses the primary somatosensory cortex and at least one other short thalamocortical pathway. 

Point 6. Please give some reference to support the dissociation between perceptual load and working memory demands suggesting some common elements across the different SAI sensorimotor circuits.

Response 6. – We apologize for any ambiguity surrounding the accidental use of “perceptual load”. It should have read “perceptual attention load.” We have also revised parts of this paragraph (lines 105 to 108) to clarify the point about common elements across the sensorimotor circuits. 

Point 7. The line number 179 is not clear.

Response 7. We have revised the text (lines 197 to 202 in the revised manuscript) to better describe electrode placement and the triggered recording of data epochs around the TMS stimulus. 

Point 8. Does PA in the verbal task reflect subtle differences in SAI and not in AP (SAI as well as LAI)?

Response 8. We apologize if we have misinterpreted the reviewer’s comment. We believe the reviewer is asking whether the specific effect of verbal working memory reflects a specific influence over the afferent pathways probed by PA-induced current. Yes, we believe that something about the working memory task is specifically changing the strength of the afferent projections to the population of interneurons preferentially recruited by PA-induced current. The effect on the afferent projection is much more robust for verbal memory than spatial memory. Finally, we believe that the dissociation between SAI and LAI to verbal working memory demands likely reflects the increasing complexity of the processing pathways across the longer interstimulus interval (200 ms) to elicit LAI. 

We hope revisions to lines 452 to 467 and lines 393 to 399 better illustrate these points. 

Point 9. The authors mentioned every critical question to be asked in this article as a limitation of this paper.

Response 9. We thank the reviewer for their comment. Our goal in the discussion was to provide a rigorous placement of our results within the current literature. Part of this process reflects the trade-off between internal and external experimental validity. While we believe our results to be valid for the dominant hand and during the window of the maintenance period under study, we felt it prudent to address the generalizability to the non-dominant hand and early memory maintenance. To make this explicit, we have revised the text spanning lines 441 to 467.

The limitation of the MagStim 2002 stimulator also attempts to provide guidance for future work that can leverage new knowledge generated by recent technological advances.

Reviewer #2 

Point 1. I think the authors should make some reference to the work using spatial working memory, as it is not clear why they have chosen this task to compare with verbal working memory.

Response 1. We thank the reviewer for their comment and recognize this oversight. We have expanded the last paragraph of the introduction (lines 124-133) to address their comment. Briefly, both verbal and spatial memory play important roles in sensorimotor control. We sought to determine if the effect of working memory on the PA and AP sensorimotor circuits was a general effect or dependent on the particular domain. 

Point 2. The order of the trials with AP and PA-induces current is not clear in the methodology. Can the authors say whether they followed an order or whether it was randomized?

Response 2. We apologize for the lack of clarity. We revised lines 246 to 262 to enhance the description of how the order of trials was determined. 

Point 3. Seeing that there are differences in the TRs, I suggest that the authors could carry out the analyses they have done by introducing the TR as a covariate, so as to shed some more light on the results obtained.

Response 3. We thank the reviewer for their comment. We are unsure what adding threshold (TR) as a covariate would do to the results. The experimental design does not have between-subject factors, so the within-subjects error term captures the covariate. 

Further, TMS threshold standardizes the effect of the TMS stimulus across participants, with the magnitude of SAI shown to be relatively consistent across stimulus intensities that elicit MEPs ranging from 0.5 to 2 microvolts (Ni et al. J Neurophysiology 2011). 

That said, we did investigate the potential effect of threshold on the SAI response to Set Size and Task using multi-level modelling to account for our experiment's repeated measures design. 

The model was set to predict six-item SAI. The model included the stimulator output required to elicit a 1 mV motor-evoked potential and two-item SAI as continuous predictors. Task (Spatial vs. Verbal) was included as a categorical factor. For PA SAI, the 1 mV TMS threshold was not a significant term in the model [F(1,20)=0.06, p=0.80]. Only the terms consistent with the Task x Set Size interaction reported in the manuscript were significant. An F-test comparing a model with the predictor 1 mV threshold against a model with all other factors excluding threshold confirmed that the factor Threshold did not significantly improve the variance explained (p=0.81). The same was also true for the effect of the TMS threshold for AP SAI, PA LAI, and AP LAI. 

Point 4. In both Figure 2 and Figure 3 there is a lot of variability in the effects between participants. Can you address this issue in the discussion?

Response 4. We thank the reviewer for their comment. For AP current, part of the variability across individuals may reflect heterogeneity in the sensorimotor circuits recruited by the fixed pulse width of the Magstim 2002 stimulator as acknowledged in the limitations section of the manuscript. In addition, we have revised the text (lines 393 to 399 and lines 452 to 467) to discuss additional sources of variability explicitly. We do not believe that the inter-individual variability compromises the group-level effects. We presented individual data in our figures to demonstrate the consistency (or inconsistency) of the effect across individuals within a task and current configuration to provide confidence in the group-level changes, with the dashed bars representing a difference from zero that would be considered meaningful (a moderate effect at the group level). For PA SAI, it is quite clear that the group effect reflects a relatively predictable decrease of meaningful magnitude across individuals not seen in other conditions. 

Point 5. Could you produce a figure showing only the participant averages for both SAI and LAI?

Response 5. We have revised figures 2 (SAI) and 3 (LAI), and their captions, to include separate figures for the group averages. As part of the revision process, we rearranged the placement of the specific figures. The task (verbal or spatial) is now arranged by row, and the current direction (PA or AP) is now arranged by column.

---

## [Decision Letter · Decision Letter 1]

8 Apr 2024

PONE-D-23-40285R1The effects of verbal and spatial working memory on short- and long-latency sensorimotor circuits in the motor cortexPLOS ONE

Dear Dr. Meehan,

Thank you for submitting your manuscript to PLOS ONE. After careful consideration, we feel that it has merit and meets PLOS ONE’s publication criteria. Yet, Reviewer 2 has a final minor point that I would like to invite you to address before making a final decision on the manuscript. 

We look forward to receiving your revised manuscript.

Kind regards,

Dimitris Voudouris

Academic Editor

PLOS ONE

Journal Requirements:

Reviewers' comments:

Reviewer's Responses to Questions

**Comments to the Author**

1. If the authors have adequately addressed your comments raised in a previous round of review and you feel that this manuscript is now acceptable for publication, you may indicate that here to bypass the “Comments to the Author” section, enter your conflict of interest statement in the “Confidential to Editor” section, and submit your "Accept" recommendation.

Reviewer #1: All comments have been addressed

Reviewer #2: All comments have been addressed

2. Is the manuscript technically sound, and do the data support the conclusions?

Reviewer #1: Partly

Reviewer #2: Yes

3. Has the statistical analysis been performed appropriately and rigorously? 

Reviewer #1: N/A

Reviewer #2: Yes

4. Have the authors made all data underlying the findings in their manuscript fully available?

Reviewer #1: Yes

Reviewer #2: Yes

5. Is the manuscript presented in an intelligible fashion and written in standard English?

Reviewer #1: Yes

Reviewer #2: Yes

6. Review Comments to the Author

**Reviewer #1**: The authors revised the manuscript "the effect of verbal and spatial working memory on short- and long-latency sensorimotor cicuits in motor cortex". And the authors incorporated my comments.

**Reviewer #2:** Thank you very much to the authors for all of your explanations and corrections. I would like to make one final suggestion. Please rewrite the two sentences contained between lines 39-42. Afferent inhibition is not a stimulation technique. Afferent inhibition is the phenomena by which a sensory afferent volley inhibits the motor response in a given muscle and is typically studied by combining non-invasive electrical nerve stimulation with TMS over M1 (Claudia V. Turco, Jenin El-Sayes, Mitchell J. Savoie, Hunter J. Fassett, Mitchell B. Locke, Aimee J. Nelson, Short- and long-latency afferent inhibition; uses, mechanisms and influencing factors, Brain Stimulation,Volume 11, Issue 1, 2018,Pages 59-74)

7. PLOS authors have the option to publish the peer review history of their article (what does this mean?). If published, this will include your full peer review and any attached files.

Reviewer #1: No

Reviewer #2: No

---

## [Author Response · Author response to Decision Letter 1]

15 Apr 2024

Response to Reviewer Comments

We thank the reviewer for their additional comment. We have made the applicable revisions. Changes made to the manuscript text are highlighted in red. In our responses, specific line numbers reference the revised manuscript. 

Reviewer #2 

Point 1. Please rewrite the two sentences contained between lines 39-42. Afferent inhibition is not a stimulation technique. Afferent inhibition is the phenomena by which a sensory afferent volley inhibits the motor response in a given muscle and is typically studied by combining non-invasive electrical nerve stimulation with TMS over M1 (Claudia V. Turco, Jenin El-Sayes, Mitchell J. Savoie, Hunter J. Fassett, Mitchell B. Locke, Aimee J. Nelson, Short- and long-latency afferent inhibition; uses, mechanisms and influencing factors, Brain Stimulation,Volume 11, Issue 1, 2018,Pages 59-74)

Response 1. We have the sentences in question (lines 39 to 43).

---

## [Editor Report · Decision Letter 2]

17 Apr 2024

The effects of verbal and spatial working memory on short- and long-latency sensorimotor circuits in the motor cortex

PONE-D-23-40285R2

Dear Dr. Meehan,

We’re pleased to inform you that your manuscript has been judged scientifically suitable for publication and will be formally accepted for publication once it meets all outstanding technical requirements.

Kind regards,

Dimitris Voudouris

Academic Editor

PLOS ONE
---

## [Editor Report · Acceptance letter]

3 May 2024

PONE-D-23-40285R2 

PLOS ONE

Dear Dr. Meehan, 

I'm pleased to inform you that your manuscript has been deemed suitable for publication in PLOS ONE. Congratulations! Your manuscript is now being handed over to our production team.

Kind regards, 

on behalf of

Dr. Dimitris Voudouris 

Academic Editor

PLOS ONE